# An Engineering Method of Analyzing the Dynamics of Mass Transfer during Concrete Corrosion Processes in Offshore Structures

**DOI:** 10.3390/ma16103705

**Published:** 2023-05-12

**Authors:** Sergey Viktorovich Fedosov, Olga Vladimirovna Aleksandrova, Azariy Abramovich Lapidus, Tatiana Konstantinovna Kuzmina, Dmitriy Vladimirovich Topchiy

**Affiliations:** Institute of Industrial and Civil Engineering, National Research Moscow State Civil Engineering University, Yaroslavskoe Shosse, House 26, 129337 Moscow, Russia; fedosovsv@mgsu.ru (S.V.F.); lapidus58@mail.ru (A.A.L.); kyzmina_tk@mail.ru (T.K.K.); 89161122142@mail.ru (D.V.T.)

**Keywords:** mass transfer, “structure–shore (soil)–water area” system, boundary problem of mass conductivity, Henry constant, conjugating boundary condition, Fourier number

## Abstract

The environment of an underground structure is much more complex than the above-ground environment. Erosion processes are underway in soil and groundwater; groundwater seepage and soil pressure are also typical for underground environments. Alternating layers of dry and wet soil have a strong effect on concrete, and they reduce its durability. Corrosion of cement concretes is caused by the diffusion of free calcium hydroxide, located in the pores of concrete, from the volume of the cement stone to its surface, bordering on an aggressive environment, and the further transition of the substance through the phase boundary solid (concrete)–soil-aggressive environment (liquid). Due to the fact that all minerals in cement stone exist only in saturated or close-to-saturated solutions of calcium hydroxide, a decrease in the content of which in the pores of concrete as a result of mass transfer processes causes a change in the phase and thermodynamic equilibrium in the body of concrete and leads to the decomposition of highly basic compounds of cement stone and, consequently, to the deterioration of the mechanical properties of concrete (reduction in strength, modulus of elasticity, etc.). A mathematical model of mass transfer in a two-layer plate imitating the “reinforced concrete structure—layer of the soil–coastal marine area” system is proposed as a system of nonstationary partial derivative differential equations of the parabolic type with Newmann’s boundary conditions inside the building and at the interface between the soil and the marine environment and with conjugating boundary conditions at the interface between the concrete and the soil. When the boundary problem of mass conductivity in the “concrete–soil” system is solved, expressions are obtained to determine the dynamics of the concentration profiles of the target component (calcium ions) in the volumes of the concrete and soil. As a result, one can select the optimum composition of concrete, having high anticorrosion properties, to extend the durability of the concrete constructions of offshore marine structures.

## 1. Introduction

Today, the construction industry is solving a high-priority task of developing effective recommendations to prevent the corrosion-induced failure of reinforced concrete structures, and the solution should be based on both classical and the most recent theoretical and experimental studies [1,2,3,4,5].

Corrosion of concrete is a complex physical and chemical process that eventually leads to the failure of reinforced concrete structures. Calcium hydroxide (CaOH) is the main component of concrete and is involved in the cement stone degradation process in the initial stage. Various mechanisms of transfer of aggressive substances to corrosive surfaces are triggered in the process of concrete corrosion [3,4,5,6,7].

Seawater is a highly aggressive medium containing large amounts of dissolved salts; it causes chemical corrosion of both concrete itself and reinforced concrete structures [8,9,10].

When analyzing the data extracted from the sources [11,12], a researcher should pay attention to the composition of the chemical components of seawater in the nearshore area. The concentration of ${\mathrm{SO}}_{4}^{2-}$ ions is 2.5–3.02 kg/m^3^ in seawater with a salt content of 33–35 kg/m^3^, which is explained by the dissolution of sulfate compounds of Ca, Na, K, and Mg, same as in fresh water, but in much larger amounts and with a predominance of sodium and magnesium salts. It is obvious that sulfate and chloride ions are the main components. Hence, we can conclude that the corrosion of concrete in structures is a complex physical and chemical process that becomes more complicated as a result of sulfate and chloride corrosion [13,14,15]. It is very important to understand that these two types of corrosion are characterized by the transfer of an aggressive component and reaction product in reciprocal directions.

The mechanism of sulfate corrosion is characterized by diffusion (mass conductivity) of free calcium hydroxide (Ca^2+^ ions) from the volume of the concrete matrix through the “concrete–soil” interface, followed by mass conductivity through the soil layer to the “soil–liquid” interface, and mass transfer into the marine medium [3,6,11,12].

Mathematical models and numerical experiments based on them enable us to evaluate the influence of certain parameters of the structure’s operation on its durability.

The mechanism of chloride corrosion is much more complicated [3,11,12]. Diffusion of chloride ions results from mass conductivity through the product layer. Moreover, Ca^2+^ ions diffuse towards chloride ions, which are transferred by means of the leaching corrosion mechanism (type I corrosion) [16,17,18,19]. Both soluble compounds like CaCl_2_ and insoluble compounds like Mg(OH)_2_ are formed in the soil layer. CaCl_2_ ions are carried into the aquatic medium, while Mg(OH)_2_ triggers pore colmatation [3,11,12]. Mass transport of active components in furtherance of the above corrosion mechanisms can be determined with differential equations of nonstationary mass conductivity (diffusion in a solid body) [20,21,22,23]. A number of factors influence the course of the concrete corrosion process, which makes it difficult to determine the dynamics of corrosion-triggered damage in concrete [24,25,26,27,28]. It is advisable to apply mathematical modeling to solve problems of this type because mathematical modeling means the use of dimensionless complexes conveying the chemical and physical essence of the phenomena under study. Such mathematical models include a system of nonlinear partial differential equations, characterizing nonstationary mass conductivity or diffusion of free calcium hydroxide in a solid body of a concrete (reinforced concrete) structure, as observed in the system of “concrete (reinforced concrete)–soil–marine environment (liquid)” in the case of nonuniform arbitrary initial conditions and in the case of combined Newmann’s boundary conditions along the outer boundaries of the system and conjugating boundary conditions along the interface between concrete and soil, taking into account the nonlinear character of mass conductivity coefficients [29,30]. The effectiveness of mathematical modeling methods as an integral part of the successful prediction of premature deterioration of structures is proven by the high rate of practical applicability of its results [31,32].

The purpose of this work is to present the results of a numerical experiment carried out according to the mathematical model of mass transfer developed by the authors in a closed system “concrete (reinforced concrete)–soil–sea environment (liquid)” for the processes of corrosion of cement stone in the presence of an internal source of mass—free calcium hydroxide in the body of concrete.

## 2. Materials and Methods

Now we consider the situation when the wall structure of an onshore facility is immersed in coastal soil at a certain distance from the shore.

Let’s assume that the values of the densities of the soil and aquatic medium are different: ρ_concrete_ ≠ ρ_soil_. The driving force of mass transfer processes is the difference in the concentrations of the transferred component at different points in space. At the boundaries of the environments “concrete–soil” and “soil–sea”, there is always an interphase equilibrium, which is determined by “Henry’s law”.

*C*_1_(*x*,*τ*); *C*_2_(*x*,*τ*) are functions determining the distributions of the concentrations of the target component in concrete and in the soil layer in the arbitrary moment of time τ.

${\hat{C}}_{A}\left(\tau \right)$ is the concentration of the component in the liquid medium of seawater, kg of Ca^2+^/kg of liquid.

$C_{1}^{*}\left(\tau \right)$, $C_{2}^{*}\left(\tau \right)$ are the concentrations of the component at the interface between the structure and soil.

Because the concentrations of the transferred component are expressed in different units, local equilibrium, as determined by “Henry’s law”, exists at the interface in question at the arbitrary moment τ [33]:(1)$$C_{1}^{*}\left(\tau \right)=m\cdot C_{2}^{*}\left(\tau \right).$$

Equation (1) is the “first part” of the conjugating boundary condition. The “second part” of the conjugating boundary condition is the condition of equality of mass flows of the transferred component that is on the way from phase “1” to phase “2”.
(2)$$-k_{1}\rho_{1}\frac{\partial C_{1}^{*}\left(\tau \right)}{\partial x}=-k_{2}\rho_{2}\frac{\partial C_{2}^{*}\left(\tau \right)}{\partial x},$$
where *k*_1_ and *k*_2_ are the coefficients of mass conductivity in the concrete and soil of the transported substance; *ρ*_1_ and *ρ*_2_ are the density of the concrete and soil, respectively.

Differential equations of unsteady diffusion, describing the dynamics of concentration fields of the transferred component (Ca^2+^) in each of the phases, have the following form [34]:(3)$$\frac{\partial C_{1}\left(x,\tau \right)}{\partial \tau }=k_{1}\frac{\partial^{2}C_{1}\left(x,\tau \right)}{\partial x^{2}};\;\tau >0;0\leq x\leq \delta_{1},$$
(4)$$\frac{\partial C_{2}\left(x,\tau \right)}{\partial \tau }=k_{1}\frac{\partial^{2}C_{2}\left(x,\tau \right)}{\partial x^{2}};\;\tau >0;\delta_{1}\leq x\leq \delta_{2}.$$

Expressions (1) and (2) determine mass transfer conditions at the interface between the contact zones of the concrete structure and the soil layer. Let us obtain the closing relations for the other sides of the interacting media.

Firstly, we assume that there is no transfer of the target component from the surface of the structure, which is the internal surface of a construction facility, into its interior. From the mathematical point of view, this means that the mass flow density of the substance is equal to zero:(5)$${\left.q_{0}\left(\tau \right)=-k_{1}\cdot \rho_{1}\frac{\partial C_{1}\left(x,\tau \right)}{\partial x}\right|}_{x=0}=0,$$
where *q*_0_ is the mass flux density through the left boundary of the considered system.

Secondly, the target component is carried from the soil of the nearshore area into the seawater according to the principle of surface mass transfer. This mass transfer is described with the equation:(6)$${\left.-k_{2}\cdot \rho_{2}\frac{\partial C_{2}\left(x,\tau \right)}{\partial x}\right|}_{x=\delta_{2}}=q_{H}\left(\tau \right),$$
(7)$$q_{H}\left(\tau \right)=\beta \left[{\hat{C}}_{A}\left(\tau \right)-m_{A}\cdot C_{2}{\left.\left(x,\tau \right)\right|}_{x=\delta_{2}}\right],$$
where *m*_*A*_ is the Henry constant for the “soil–water area” system, and *β* is the coefficient of mass transfer from the soil to the water area.

It is very important to understand that the system of Equations (3) and (4) has a single solution if supplemented with marginal conditions. Marginal conditions include boundary conditions, as well as initial conditions determining fields of concentrations of a transferred component at the moment in time taken as a starting point, or τ_0_.

According to Figure 1, marginal conditions can be represented as the following dependences:(8)$$C_{1}(x,\tau {\left.)\right|}_{\tau =0}=C_{1.0}\left(x\right),$$
(9)$$C_{2}(x,\tau {\left.)\right|}_{\tau =0}=C_{2.0}\left(x\right)\;$$

Diffusion mass transfer begins at τ_0_. The conjugating boundary condition develops at the interface between the solid phases:(10)$$-k_{1}{\left.\rho_{1}\frac{\partial C_{1}\left(x,\tau \right)}{\partial x}\right|}_{x=\delta_{1}}=-k_{2}{\left.\rho_{2}\frac{\partial C_{2}\left(x,\tau \right)}{\partial x}\right|}_{x=\delta_{1}},$$
(11)$$C_{1}(x,\tau {\left.)\right|}_{x=\delta_{1}\;}=mC_{2}(x,\tau {\left.)\right|}_{x=\delta_{1}\;}.\;$$

The boundary problem of mass conductivity in a two-layer plate is addressed and solved in [11,12,35]. The general statement of the nonstationary mass conductivity problem for an unlimited model plate can be formulated as follows:-for an enclosing structure:


(12)
$$\frac{\partial C_{1}\left(x,\tau \right)}{\partial \tau }=k_{1}\;\frac{\partial^{2}C_{1}\left(x,\tau \right)}{\partial x^{2}};\;0\leq x\leq \delta_{1},$$



(13)
$$C_{1}(x,\tau {\left.)\right|}_{\tau =0\;}=C_{1.0}\left(x\right),$$



(14)
$$q_{0}\left(\tau \right)={\left.-k_{1}\rho_{1}\frac{\partial C_{1}\left(x,\tau \right)}{\partial x}\right|}_{x=0},$$



(15)
$$q_{1}\left(\tau \right)=-k_{1}{\left.\rho_{1}\frac{\partial C_{1}\left(x,\tau \right)}{\partial x}\right|}_{x=\delta_{1}},$$


-for a soil layer:

(16)$$\frac{\partial C_{2}\left(x,\tau \right)}{\partial \tau }=k_{2}\;\frac{\partial^{2}C_{2}\left(x,\tau \right)}{\partial x^{2}};\;\;\delta_{1}\leq x\leq \delta_{2}\;,$$(17)$$C_{2}(x,\tau {\left.)\right|}_{\tau =0\;}=C_{2.0}\left(x\right),$$(18)$$q_{2}\left(\tau \right)={\left.-k_{2\;\;}\rho_{2}\frac{\partial C_{2}\left(x,\tau \right)}{\partial x}\right|}_{x=\delta_{1}},$$(19)$$q_{A}\left(\tau \right)=-{\left.k_{2}\rho_{2}\frac{\partial C_{2}\left(x,\tau \right)}{\partial x}\right|}_{x=\delta_{2}},$$
where *q*_1_ is the density of the mass flow from the structure to the boundary with the soil; *q*_2_ is the mass flow from the soil boundary; *q*_*A*_ is the density of the mass flow from the soil to the sea area.

Hence, from the standpoint of formal logic, a solution to marginal problems (12)–(15) and (16)–(19) can be provided in the following form [3,34,36,37,38]:


(20)
$$\begin{array}{ll} C_{1}\left(x,\tau \right) & =\frac{1}{\delta_{1}}\overset{\delta_{1}}{\underset{0}{\int }}C_{1.0}\left(x\right)\cdot dx-\frac{1}{\delta_{1}}\overset{\tau }{\underset{0}{\int }}\left[q_{0}\left(\tau^{*}\right)-q_{1}\left(\tau^{*}\right)\right]d\tau^{*}\\ & +\frac{2}{\delta_{1}}\overset{\infty }{\underset{m=1}{\sum }}\cos\left(\pi m\bar{x}\right)\cdot \exp\left(-\pi^{2}m^{2}Fo_{1}\right)\cdot \overset{\delta_{1}}{\underset{0}{\int }}C_{1.0}\left(x^{*}\right)\cos\left(\pi m\xi^{*}\right)dx^{*}\\ & -\frac{2}{\delta_{1}}\overset{\infty }{\underset{m=1}{\sum }}\cos\left(\pi m\bar{x}\right)\cdot \overset{\tau }{\underset{0}{\int }}\left[q_{0}\left(\tau^{*}\right)-{\left(-1\right)}^{m}q_{1}\left(\tau^{*}\right)\right]\\ & \cdot \exp\left[-\pi^{2}m^{2}\left(Fo_{m.1}-Fo_{m.1}^{*}\right)\right]d\tau^{*}.\; \end{array}$$


Here, *x** is a coordinate in the construction in the range [0,*x*]; *τ** is time in the range [0,*τ*]; $\bar{x}=x/\delta_{1}$ is a dimensionless coordinate in the range [0,1]; $\xi^{*}=x^{*}/\delta_{1}$ is a dimensionless coordinate in the range [0,$\bar{x}$];
(21)$$Fo_{m.1}=\frac{k_{1}\tau }{\delta_{1}^{2}};\;Fo_{m.1}^{*}=\frac{k_{1}\left(\tau -\tau^{*}\right)}{\delta_{1}^{2}},$$
(22)$$C_{2}\left(x,\tau \right)=\frac{1}{\delta_{2}}\int_{\delta_{1}}^{\delta_{2}}C_{2.0}\left(x\right)\cdot dx-\frac{1}{\delta_{2}}\int_{0}^{\tau }\left[q_{A}\left(\tau^{*}\right)-q_{2}\left(\tau^{*}\right)\right]d\tau^{*}+\;\frac{2}{\delta_{2}}\sum_{m=1}^{\infty }\cos\left(\pi m\overset{=}{x}\right)\cdot \exp\left(-\pi^{2}m^{2}Fo_{m.2}\right)\int_{\delta_{1}}^{\delta_{2}}C_{2.0}\left(x^{**}\right)\cos\left(\pi m\xi^{**}\right)dx^{**}-\;\frac{2}{\delta_{2}}\sum_{m=1}^{\infty }\cos\left(\pi m\overset{=}{x}\right)\cdot \int_{0}^{\tau }\left[q_{2}\left(\tau^{*}\right)-{\left(-1\right)}^{m}q_{A}\left(\tau^{*}\right)\right]\cdot \exp\left[-\pi^{2}m^{2}\left(Fo_{m.2}-\;Fo_{m.2}^{*}\right)\right]d\tau^{*},$$
(23)$$\overset{=}{x}=\frac{x}{\delta_{2}};\;\;\xi^{**}=\frac{x^{**}}{\delta_{2}};\;\;Fo_{m.2}=\frac{k_{2}\tau }{\delta_{2}^{2}}\;;\;Fo_{m.2}^{*}=\frac{k_{2}\left(\tau -\tau^{*}\right)}{\delta_{2}^{2}}.$$

Here, *x*** is the coordinate in the ground in the range [0,*x*].

Now we address the problem of boundary conditions. A condition on the inner surface of a structure means that no transferable component travels from the structure to the inner space. Mathematically, this condition is written as follows:(24)$${\left.\frac{\partial C_{1}\left(x,\tau \right)}{\partial x}\right|}_{x=0}=0,$$
and this is the “impermeability condition”.

The proposed methodology for solving the mass transfer problems under study involves the use of the “microprocess method” in calculations [31,39,40]. This method assumes the constancy of transfer coefficients during the time of the microprocess. Hence, the first boundary condition (on the left inner boundary) will have the following form:(25)$$q_{0}\left(\tau \right)=\mathrm{const}=0.$$

The conjugating boundary condition in the place of contact between the solid phases (the structure and soil) is determined by the conditions of expressions (1) and (2), or (10) and (11).

The boundary condition in the place of contact between the coastline soil and seawater is determined with expressions (6) and (7), provided that:(26)$$q_{H}\left(\tau \right)=q_{A}\left(\tau \right).$$

Here, *q*_*H*_(*τ*) is mass flux density as determined by Equation (7).

Experimental information is needed to initiate the use of the engineering calculation method:Initial values of concentrations.Values of Henry constants “*m*” as the equilibrium in the “structure–soil” system.Values of Henry constants “*m*_A_” as the equilibrium in the “soil–water area” system.Value of coefficient “*β*” that shows mass transfer from the soil to the water area.

The calculation method is implemented in compliance with the following algorithm: First, we take into account that the calculation starts in the case of uniform initial concentration distributions, i.e., the following items are set:(27)$$C_{1}(x,\tau {\left.)\right|}_{\tau =0\;}=C_{1.0\;},$$
(28)$$C_{2}(x,\tau {\left.)\right|}_{\tau =0\;}=C_{2.0}.$$

Well-known reference data are sourced for the values of the equilibrium constants in the “reinforced concrete–soil” system: m [(kg of Ca^+2^/kg of concrete)/(kg of Ca^+2^/kg of soil)].

These conditions correspond to the straight lines in Figure 1 and Figure 2: *C*_1.0_ = const; *C*_2.0_= const.

Assuming such initial conditions, we should also make an assumption that the process of mass transfer from the structure to the soil is also nonstationary during the initial period before the attainment of equilibrium at the interface.

In this case, the time of attainment of this equilibrium,$\;\tau_{p}^{*}$, is determined by the geometrical characteristics of the “structure–soil” system and mass transfer properties of the substances: *k*_1_, *k*_2_.

In the first stage of the calculation:

(1a) The concentration change in the structure is arbitrarily selected at the interface ∆C_1_, and the corresponding concentration value is determined:(29)$$C_{1.0}^{*}=C_{1.0}-\Delta C_{1};$$

(1b) The calculation step along coordinate *x* → ∆*x*_1_ is selected;

(1c) The value of the concentration gradient is determined:(30)$$\nabla C_{1}^{*}\left(\tau_{1}\right)=gradC_{1}^{*}\left(\tau_{1}\right)=\frac{\Delta C_{1}^{*}}{\Delta x_{1}};$$

(1d) The value of the substance flow density is determined:(31)$$q_{1}\left(\tau_{1}\right)=-k_{1}\rho_{1}\underset{\Delta x\to 0}{\lim}\frac{\Delta C_{1}^{*}}{\Delta x_{1}}=-k_{1}\rho_{1}\nabla C_{1}^{*}\left(\tau_{1}\right)=-k_{1}\rho_{1}\frac{dC_{1}^{*}\left(\tau_{1}\right)}{dx};$$

(1e) The value of the time interval ∆τ_1_ is selected, and the corresponding value of the Fourier number is determined:(32)$$Fo_{m.1}=\frac{k_{1}\cdot \Delta \tau_{1}}{\delta_{1}^{2}};$$

(1f) The distribution of the concentrations over the thickness of the structure is calculated, and the value of $C_{1}^{*}\left(\tau_{1}\right)$ is found using Equation (20). This calculated value is compared with the pre-set one:(33)$$C_{1.0}^{*}-C_{1}^{*}\left(\tau_{1}\right)=\varepsilon_{1},$$
and if error *ε*_1_ is unsatisfactory, the calculation process restarts from position (1a).

If the condition is satisfied, the next stage of the calculation is initiated.

In the second stage of the calculation:

(2a) The boundary condition for the problem of concentration distribution in the soil layer is found using boundary condition (1d):(34)$$q_{2}\left(\tau_{1}\right)=-k_{2}\rho_{2}\frac{dC_{2}^{*}\left(\tau_{1}\right)}{dx};$$

(2b) We know the coefficient of mass flow in the liquid phase that equals *β* and the Henry coefficient in the “soil–liquid” system that equals *m_A_*.

The task is to find the value of mass flow density for the external boundary condition (7);

(2c) Below is the calculation of the concentration distribution in the soil layer for the moment $\tau_{1}^{*}$ and the corresponding Fourier criterion:(35)$$Fo_{m.1}\left(\tau_{1}^{*}\right)=\frac{k_{2}\cdot \Delta \tau_{1}}{\delta_{2}^{2}},$$
and the same calculation also identifies the value of the concentration in the soil at the boundary of the structure $\;C_{2}^{*}\left(\tau_{1}^{*}\right)$.

Figure 3 illustrates the calculation algorithm.

The obtained values of concentrations are compared:(36)$$C_{1}^{*}\left(\tau_{1}\right)-mC_{2}^{*}\left(\tau_{1}\right)=\varepsilon_{2},$$
and if the pre-set value ε_2_ is not satisfied, a new value $\Delta C_{1}(\tau_{2})$ is selected, and the calculation process is repeated. The total process time is summed up.

In the third step, when equilibrium is attained at the interface between the solid phases, the process of mass transfer continues. This process is mass transfer from the reinforced concrete structure through the soil layer into the sea area.

## 3. Results and Discussion

Numerical experiments were conducted to evaluate the adequacy of the proposed mathematical model, which takes into account the extent of influence of soil and seawater salinity properties on free calcium hydroxide mass transfer processes that accompany corrosion-induced damage to concrete in the liquid medium.

Let us transform Equations (20) and (22).

Let us write down the dimensionless concentration:(37)$$U_{1}\left(\bar{x},Fo_{m.1}\right)=\frac{C_{1}\left(x,\tau \right)}{C_{1.0}^{*}},$$
where $C_{1.0}^{*}$ is the value of concentration of the transferred component (Ca^2+^) in the initial moment at the point having the coordinate *x* = 0.

Dimensionless constant: $\bar{x}=\frac{x}{\delta_{1}}$
(38)$$Fo_{m.1}=\frac{k_{1}\tau_{1}}{\delta_{1}^{2}};\;Fo_{m.1}^{*}=\frac{k_{1}\left(\tau -\tau^{*}\right)}{\delta_{1}^{2}}.\;$$

Hence, by transforming (20), we write down the following:


(39)
$$\begin{array}{ll} U_{1}\left(\bar{x},Fo_{m.1}\right) & =\frac{1}{\delta_{1}}\overset{\delta_{1}}{\underset{0}{\int }}\frac{C_{1.0}\left(x\right)}{C_{1.0}^{*}}\cdot dx-\frac{1}{\delta_{1}}\overset{\tau }{\underset{0}{\int }}\left[\frac{q_{0}\left(\tau^{*}\right)}{C_{1.0}^{*}}-\frac{q_{1}\left(\tau^{*}\right)}{C_{1.0}^{*}}\right]\cdot d\tau^{*}\\ & +\frac{2}{\delta_{1}}\overset{\infty }{\underset{m=1}{\sum }}cos(\pi m\bar{x})\cdot exp\left(-\pi^{2}m^{2}Fo_{m.1}\right)\cdot \overset{\delta_{1}}{\underset{0}{\int }}\frac{C_{1.0}\left(x^{*}\right)}{C_{1.0}^{*}}\\ & \cdot cos\left(\pi m\xi^{*}\right)\cdot dx^{*}-\frac{2}{\delta_{1}}\overset{\infty }{\underset{m=1}{\sum }}cos(\pi m\bar{x})\\ & \cdot \overset{\tau }{\underset{0}{\int }}\left[\frac{q_{0}\left(\tau^{*}\right)}{C_{1.0}^{*}}-{\left(-1\right)}^{m}\cdot \frac{q_{1}\left(\tau^{*}\right)}{C_{1.0}^{*}}\right]\\ & \cdot exp\left[-\pi^{2}m^{2}\left(Fo_{m.1}-Fo_{m.1}^{*}\right)\right]d\tau^{*}.\; \end{array}$$


Let us transform each term of the right-hand side of Equation (39) one by one.

The first summand is as follows:(40)$$\frac{1}{\delta_{1}}\int_{0}^{\delta_{1}}\frac{C_{1.0}\left(x\right)}{C_{1.0}^{*}}\cdot dx=\frac{1}{\delta_{1}}\int_{0}^{\delta_{1}}U_{1.0}\left(x\right)\cdot d\left(x\right)=\int_{0}^{1}U_{1.0}\left(\bar{x}\right)\cdot d\left(\frac{x}{\delta_{1}}\right)=\int_{0}^{1}U_{1.0}\left(\bar{x}\right)\cdot d\bar{x}.$$

Let us add a complex to the second summand: $\frac{k_{1}}{k_{1}}\cdot \frac{\delta_{1}}{\delta_{1}}$
(41)$$\frac{1}{\delta_{1}}\int_{0}^{\tau }\left[\frac{q_{0}\left(\tau^{*}\right)}{C_{1.0}^{*}}-\frac{q_{1}\left(\tau^{*}\right)}{C_{1.0}^{*}}\right]\cdot d\tau^{*}=-\frac{1}{\delta_{1}}\cdot \frac{k_{1}}{k_{1}}\cdot \frac{\delta_{1}}{\delta_{1}}\int_{0}^{\tau }\left[\frac{q_{0}\left(\tau^{*}\right)}{C_{1.0}^{*}}-\frac{q_{1}\left(\tau^{*}\right)}{C_{1.0}^{*}}\right]\cdot d\tau^{*}=\;-\int_{0}^{\tau }\left[\frac{q_{0}\left(\tau^{*}\right)\cdot \delta_{1}}{k_{1}\cdot C_{1.0}^{*}}-\frac{q_{1}\left(\tau^{*}\right)\cdot \delta_{1}}{k_{1}\cdot C_{1.0}^{*}}\right]\cdot d\left(\frac{k_{1}\tau^{*}}{\delta_{1}^{2}}\right)=-\int_{0}^{Fo_{m.1}}\left[Ki_{m.0}\left(Fo_{m.1}^{*}\right)-\;Ki_{m.1}\left(Fo_{m.1}^{*}\right)\right]dFo_{m.1}^{*},$$
(42)$$Ki_{m.0}\left(Fo_{m.1}^{*}\right)=\frac{q_{0}\left(\tau^{*}\right)\cdot \delta_{1}}{k_{1}\cdot C_{1.0}^{*}};\;Ki_{m.1}\left(Fo_{m.1}^{*}\right)=\frac{q_{1}\left(\tau^{*}\right)\cdot \delta_{1}}{k_{1}\cdot C_{1.0}^{*}};\;Fo_{m.1}^{*}=\frac{k_{1}\tau^{*}}{\delta_{1}^{2}}.$$

The third summand (39) is easily transformed by introducing *δ*_1_ under the sign of the differential:(43)$$\frac{2}{\delta_{1}}\sum_{m=1}^{\infty }cos(\pi m\bar{x})\cdot exp\left(-\pi^{2}m^{2}Fo_{m.1}\right)\cdot \int_{0}^{\delta_{1}}\frac{C_{1.0}\left(x^{*}\right)}{C_{1.0}^{*}}\cdot \cos\left(\pi m\xi^{*}\right)\cdot dx^{*}=\;+2\sum_{m=1}^{\infty }cos(\pi m\bar{x})\cdot exp\left(-\pi^{2}m^{2}Fo_{m.1}\right)\cdot \int_{0}^{1}U_{1.0}\left(\xi^{*}\right)\cdot cos\left(\pi m\xi^{*}\right)d\xi^{*}\;.$$

The fourth summand, like the second one, is transformed by its multiplying by the complex $\frac{k_{1}}{k_{1}}\cdot \frac{\delta_{1}}{\delta_{1}}$
(44)$$-\frac{k_{1}}{k_{1}}\cdot \frac{\delta_{1}}{\delta_{1}}\frac{2}{\delta_{1}}\sum_{m=1}^{\infty }cos(\pi m\bar{x})\cdot \int_{0}^{\tau }\left[\frac{q_{0}\left(\tau^{*}\right)}{C_{1.0}^{*}}-{\left(-1\right)}^{m}\frac{q_{1}\left(\tau^{*}\right)}{C_{1.0}^{*}}\right]\cdot exp\left[-\pi m\left(Fo_{m.1}-\;Fo_{m.1}^{*}\right)\right]d\tau^{*}=-2\sum_{m=1}^{\infty }cos(\pi m\bar{x})\cdot \int_{0}^{Fo_{m.1}}\left[Ki_{m.0}\left(Fo_{m.1}^{*}\right)-{\left(-1\right)}^{m}\cdot \;Ki_{m.1}\left(Fo_{m.1}^{*}\right)\right]exp\left[-\pi^{2}m^{2}\left(Fo_{m.1}-Fo_{m.1}^{*}\right)\right]dFo_{m.1}^{*}$$

Thus, the general final solution (20) is as follows:(45)$$U_{1}\left(\bar{x},Fo_{m.1}\right)=\int_{0}^{1}U_{1.0}\left(\bar{x}\right)\cdot d\bar{x}-\int_{0}^{Fo_{m.1}}\left[Ki_{m.0}\left(Fo_{m.1}^{*}\right)-Ki_{m.1}\left(Fo_{m.1}^{*}\right)\right]dFo_{m.1}^{*}+\;2\sum_{m=1}^{\infty }cos(\pi m\bar{x})\cdot exp\left(-\pi^{2}m^{2}Fo_{m.1}\right)\cdot \int_{0}^{1}U_{1.0}\left(\xi^{*}\right)\cdot \cos\left(\pi m\xi^{*}\right)d\xi^{*}-\;2\sum_{m=1}^{\infty }cos(\pi m\bar{x})\cdot \int_{0}^{Fo_{m.1}}\left[Ki_{m.0}\left(Fo_{m.1}^{*}\right)-{\left(-1\right)}^{m}\cdot \;Ki_{m.1}\left(Fo_{m.1}^{*}\right)\right]exp\left[-\pi^{2}m^{2}\left(Fo_{m.1}-Fo_{m.1}^{*}\right)\right]dFo_{m.1}^{*}.$$

We write down solution (22) and make the same transformations but omit the mathematical calculations:(46)$$\begin{array}{ll} U_{2}\left(\overset{=}{x},Fo_{m.2}\right) & =\frac{C_{2}\left(x,\tau \right)}{C_{2.0}^{*}}\\ & =\int_{0}^{1}U_{2.0}\left(\overset{=}{x}\right)\cdot d\overset{=}{x}\\ & -\int_{0}^{Fo_{m.2}}\left[Ki_{m.2}\left(Fo_{m.2}^{*}\right)-Ki_{m.A}\left(Fo_{m.2}^{*}\right)\right]dFo_{m.2}^{*}\\ & +2\overset{\infty }{\underset{m=1}{\sum }}cos(\pi m\overset{=}{x})\cdot exp\left(-\pi^{2}m^{2}Fo_{m.2}\right)\\ & \cdot \int_{0}^{1}U_{2.0}\left(\xi^{**}\right)\cdot \cos\left(\pi m\xi^{**}\right)d\xi^{**}-2\overset{\infty }{\underset{m=1}{\sum }}cos(\pi m\overset{=}{x})\\ & \cdot \int_{0}^{Fo_{m.2}}\left[Ki_{m.2}\left(Fo_{m.2}^{*}\right)-{\left(-1\right)}^{m}\cdot Ki_{m.A}\left(Fo_{m.2}^{*}\right)\right]\\ & \cdot exp\left[-\pi^{2}m^{2}\left(Fo_{m.2}-Fo_{m.2}^{*}\right)\right]dFo_{m.2}^{*}. \end{array}$$

Next, we obtain solutions for the two important special cases:Uniform initial distribution;Constancy of Kirpichev criteria values.

As for the first special case (a), solutions are easily obtained from expressions (45) and (46):(47)$$\int_{0}^{1}U_{1.0}\left(\bar{x}\right)\cdot d\bar{x}\;U_{1.0};\;\int_{0}^{1}U_{2.0}\left(\overset{=}{x}\right)\cdot d\overset{=}{x}=U_{2.0}.$$

As for the second case, the solution can be found using the transformation of expression (45):(48)$$U_{1}\left(\bar{x},Fo_{m.1}\right)==\int_{0}^{1}U_{1.0}\left(\bar{x}\right)d\bar{x}-\left[Ki_{m.0}-Ki_{m.1}\right]Fo_{m.1}+2\sum_{m=1}^{\infty }cos(\pi m\bar{x})\cdot \;exp\left(-\pi^{2}m^{2}Fo_{m.1}\right)\cdot \int_{0}^{1}U_{1.0}\left(\xi^{*}\right)\cdot \cos\left(\pi m\xi^{*}\right)d\xi^{*}-2\sum_{m=1}^{\infty }cos(\pi m\bar{x})\cdot \;\left[Ki_{m.0}-{\left(-1\right)}^{m}\cdot Ki_{m.1}\right]\cdot \int_{0}^{Fo_{m.1\;}}exp\left[-\pi^{2}m^{2}\left(Fo_{m.1}-Fo_{m.1}^{*}\right)\right]dFo_{m.1}^{*}.$$

The integral in the final summand is calculated as follows:(49)$$\int_{0}^{Fo_{m.1}}exp\left[-\pi^{2}m^{2}\left(Fo_{m.1}-Fo_{m.1}^{*}\right)\right]dFo_{m.1}^{*}=\int_{0}^{Fo_{m.1.}}exp\left[-\pi^{2}m^{2}Fo_{m.1}\right]\cdot \;exp\left[\pi^{2}m^{2}Fo_{m.1}^{*}\right]dFo_{m.1}^{*}=exp\left[-\pi^{2}m^{2}Fo_{m.1}\right]\cdot \;\int_{0}^{Fo_{m.1}}exp\left[\pi^{2}m^{2}Fo_{m.1}^{*}\right]dFo_{m.1}^{*}=exp\left[-\pi^{2}m^{2}Fo_{m.1}\right]\cdot \frac{1}{\pi m}\cdot \;e{\left.xp\left[\pi^{2}m^{2}Fo_{m.1}^{*}\right]\right|}_{0}^{Fo_{m.1}}=\frac{1}{\pi m}\cdot exp\left[-\pi^{2}m^{2}Fo_{m.1}\right]\cdot \left[exp\left(\pi^{2}m^{2}Fo_{m.1}\right)-\;exp\left(0\right)\right]=\frac{1}{\pi m}\left[1-exp\left(-\pi^{2}m^{2}Fo_{m.1}\right)\right],$$
and then by substituting the integral into (48), we obtain:(50)$$\begin{array}{ll} U_{1}\left(\bar{x},Fo_{m.1}\right)\frac{1}{\pi m} & \left[1-exp\left(-\pi^{2}m^{2}Fo_{m.1}\right)\right]\\ & =\overset{1}{\underset{0}{\int }}U_{1.0}\left(\bar{x}\right)d\bar{x}-\left[Ki_{m.0}-Ki_{m.1}\right]Fo_{m.1}+2\overset{\infty }{\underset{m=1}{\sum }}cos(\pi m\bar{x})\\ & \cdot exp\left(-\pi^{2}m^{2}Fo_{m.1}\right)\cdot \overset{1}{\underset{0}{\int }}U_{1.0}\left(\xi^{*}\right)\cdot \cos\left(\pi m\xi^{*}\right)d\xi^{*}\\ & -\frac{2}{\pi }\overset{\infty }{\underset{m=1}{\sum }}\frac{1}{m}\cdot cos(\pi m\bar{x})\cdot \left[Ki_{m.0}-{\left(-1\right)}^{m}\cdot Ki_{m.1}\right]\\ & \cdot \left[1-exp\left(-\pi^{2}m^{2}Fo_{m.1}\right)\right]. \end{array}$$

Similarly, the transformation of Equation (46) leads to the result:(51)$$\begin{array}{ll} U_{2}\left(\overset{=}{x},Fo_{m.2}\right) & =\frac{C_{2}\left(x,\tau \right)}{C_{2.0}^{*}}\\ & =\int_{0}^{1}U_{2.0}\left(\overset{=}{x}\right)\cdot d\overset{=}{x}-\left[Ki_{m.2}-Ki_{m.A}\right]dFo_{m.2}^{*}+2\overset{\infty }{\underset{m=1}{\sum }}cos(\pi m\overset{=}{x})\\ & \cdot exp\left(-\pi^{2}m^{2}Fo_{m.2}\right)\cdot \int_{0}^{1}U_{2.0}\left(\xi^{**}\right)\cdot \cos\left(\pi m\xi^{**}\right)d\xi^{**}\\ & -\frac{2}{\pi }\overset{\infty }{\underset{m=1}{\sum }}\frac{1}{m}cos(\pi m\overset{=}{x})\cdot \left[Ki_{m.2}-{\left(-1\right)}^{m}\cdot Ki_{m.A}\right]\\ & \cdot \left[1-exp\left(-\pi^{2}m^{2}Fo_{m.2}\right)\right]. \end{array}$$

Hence, these are the solutions for the constant values of the Kirpichev criteria.

Finally, for the case of the uniform initial distribution of concentrations and constant values of the Kirpichev criteria, we obtain:(52)$${\left.\frac{\partial C_{1}\left(x,\tau \right)}{\partial x}\right|}_{x=0}=0\to \;Ki_{m.0}=0,$$
and then expression (50) becomes even simpler:(53)$$\begin{array}{ll} U_{1}\left(\bar{x},Fo_{m.1}\right) & =\int_{0}^{1}U_{1.0}\left(\bar{x}\right)d\bar{x}+Ki_{m.1}\cdot Fo_{m.1}+2\overset{\infty }{\underset{m=1}{\sum }}cos(\pi m\bar{x})\\ & \cdot exp\left(-\pi^{2}m^{2}Fo_{m.1}\right)\cdot \int_{0}^{1}U_{1.0}\left(\xi^{*}\right)\cdot \cos\left(\pi m\xi^{*}\right)d\xi^{*}\\ & +\frac{2}{\pi }\overset{\infty }{\underset{m=1}{\sum }}\frac{{\left(-1\right)}^{m}}{m}\cdot cos(\pi m\bar{x})\cdot Ki_{m.1}\left[1-exp\left(-\pi^{2}m^{2}Fo_{m.1}\right)\right]. \end{array}$$

The final solutions to problems of nonstationary mass conductivity in dimensionless form, applied to the special case of uniform initial concentration distributions and constant values of the Kirpichev criteria, are as follows:(54)$$U_{1}\left(\bar{x},Fo_{m.1}\right)=U_{1.0}+Ki_{m.1}\cdot Fo_{m.1}+\frac{2}{\pi }\sum_{m=1}^{\infty }\frac{{\left(-1\right)}^{m}}{m}\cdot cos(\pi m\bar{x})\cdot Ki_{m.1}\left[1-\;exp\left(-\pi^{2}m^{2}Fo_{m.1}\right)\right],$$
(55)$$U_{2}\left(\overset{=}{x},Fo_{m.2}\right)=U_{2.0}-\left[Ki_{m.2}-Ki_{m.A}\right]Fo_{m.2}^{*}-\frac{2}{\pi }\sum_{m=1}^{\infty }\frac{1}{m}cos(\pi m\overset{=}{x})\cdot \left[Ki_{m.2}-\;{\left(-1\right)}^{m}\cdot Ki_{m.A}\right]\cdot \left[1-exp\left(-\pi^{2}m^{2}Fo_{m.2}\right)\right].$$

Some of the results of the calculations made in (54) and (55) are shown in Figure 4 and Figure 5 as graphic dependences illustrating the dynamics of dimensionless concentration fields of the transferred component along the dimensionless coordinate “structure–soil”. All lines in these figures show the influence of the Kirpichev criteria on the nature of the mass exchange process. In its physical sense, the Kirpichev criterion (in Equation (42)) characterizes the ratio of mass flows of the substance approaching the surface of the structure (or soil) from the external medium (or in the opposite direction) to the mass flow of the substance brought from the inside of the solid phase by mass conductivity.

Figure 4 shows the concentration profiles in the structure in the absence of the mass flow in the plane x = 0 (Ki_m.1_ = 0). Obviously, the greater the value of the Kirpichev number at the boundary with the soil, the greater the value of the tangent angle of the lines at the “structure–soil” boundary.

The data in Figure 5 illustrate the dynamics of the fields of dimensionless concentrations in the system “reinforced concrete structures–soil–sea area”. Calculations were performed for different values of initial concentrations. It is assumed that in the initial moment of time τ = τ_0_, the transferred component is saturated in the material of the structure:(56)$$U_{1.0}{\left.\left(\bar{x},Fo_{m.1}\;\right)\right|}_{Fo_{m.1}=0}=1,$$
and at the same time, the content of Ca^2+^ in the soil is close to zero:(57)$$U_{2.0}{\left.\left(\overset{=}{x,}Fo_{m.2}\;\right)\right|}_{Fo_{m.2}=0}=0.$$

The calculated dynamic profiles of concentrations and process kinetics make it possible to solve both direct and inverse problems of mass conductivity. The direct problem consists in calculating the dynamics of the concentration fields of a transferred component in the simulated system during the process time.

The objective of the inverse problem is to use experimental and numerical methods to determine the kinetic coefficients of the process: mass conductivity *k*_1_ and *k*_2_, and the coefficient of mass transfer in liquid phase *β*.

In a reinforced concrete structure, the change in the concentration of the transferred component occurs from the maximum content of U_0_ to the working concentration (Figure 5a); in the soil, the initial content of the transferred component is minimal, and in the process of interaction with the structure, it increases in thickness but decreases as a result of the diffusion of the right boundary into the sea (Figure 5b).

## 4. Conclusions

The above expressions facilitate the determination of the value of concentrations of the transferred component over the thickness of the concrete structure and soil at any time, and they also enable the calculation of the concentration of free calcium hydroxide in the liquid phase, calculating the process kinetics according to solid phases in concrete and soil, which ultimately enables the forecasting of the durability and reliability of building structures with a minimum error.

In addition, the calculated values of the concentration profiles make it possible to find the maximum values of the concentration gradients and their location. Accordingly, the concentration gradients determine the stresses that can occur in the structure and which lead to the appearance of microcracks, which are the cause of a decrease in the strength of the concrete structure.

## Figures and Tables

**Figure 1 materials-16-03705-f001:**
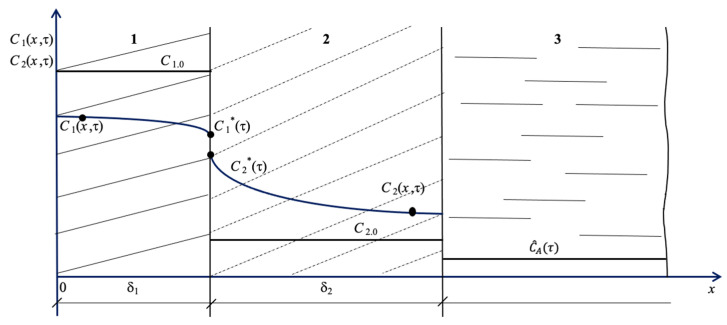
The “structure–shore (soil)–water area” model system, where 1 is the structure; 2 is soil; 3 is the seawater area; δ_1_ is the size of the structure, m; δ_2_ is the size (depth) of coastal soil, m; *C*_1.0_ is the initial concentration of the transferred component in the reinforced concrete structure, kg of Ca^2+^/kg of concrete; and *C*_2.0_ is the initial concentration of the transferred component, kg of Ca^2+^/kg of soil.

**Figure 2 materials-16-03705-f002:**
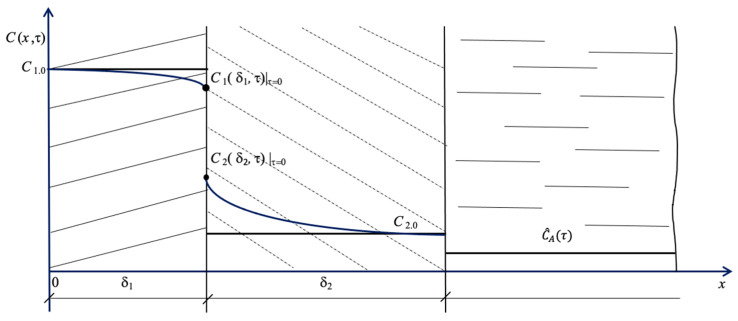
Illustration of initial distributions of Ca^2+^ concentrations in the system under study.

**Figure 3 materials-16-03705-f003:**
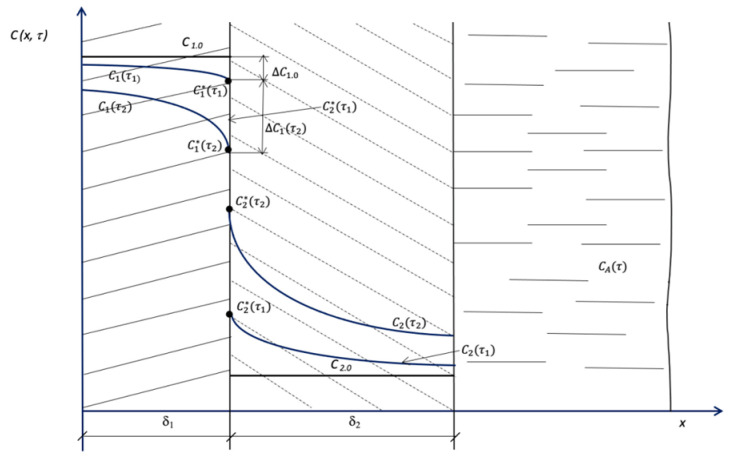
Illustration of calculations.

**Figure 4 materials-16-03705-f004:**
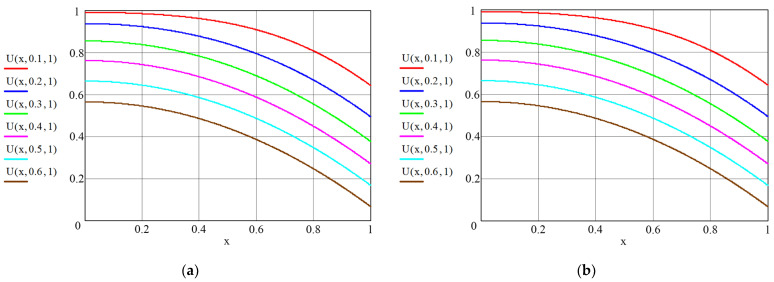
Profiles of dimensionless concentrations over the thickness of concrete: (**a**) At ${\mathrm{Ki}}_{m.1}$ = 0, at ${\mathrm{Ki}}_{m.2}$ = 0.5; (**b**) At ${\mathrm{Ki}}_{m.1}$ = 0, at ${\mathrm{Ki}}_{m.2}$ = 1.

**Figure 5 materials-16-03705-f005:**
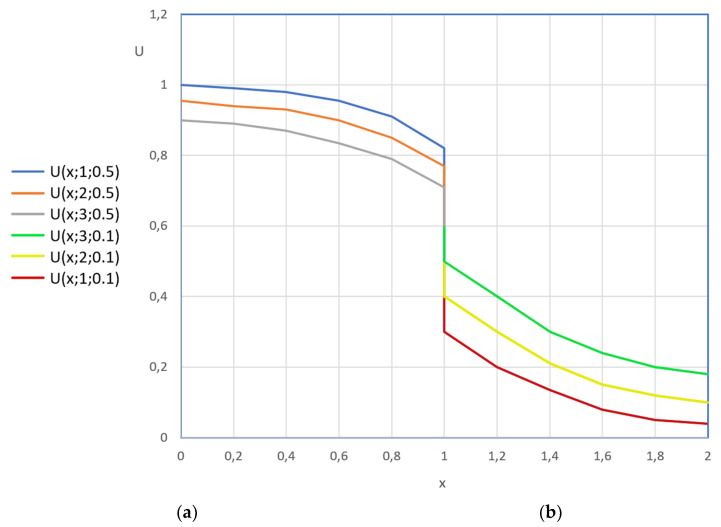
Profiles of dimensionless concentrations in the “structure–soil” system: (**a**) concrete at ${\mathrm{Ki}}_{m.1}$ = 0, at ${\mathrm{Ki}}_{m.2}$ = 0.5; (**b**) soil at ${\mathrm{Ki}}_{m.1}$ = 0.5, at ${\mathrm{Ki}}_{m.2}$ = 0.1.

## Data Availability

Data is available upon request from the corresponding author.

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
