# Peer review of "An Engineering Method of Analyzing the Dynamics of Mass Transfer during Concrete Corrosion Processes in Offshore Structures"

_materials, 2023, doi:10.3390/ma16103705_

Round 1

Reviewer 1 Report

1.  Please state reason why you conduct the research in the abstract?

2. Please use the relavent source of equations.

3. Symbol used in the equation need to be explained.

4. No validation for the study. It would be great to compare similar findings.

5. Please justify the selection of parameter in your study and why it is used in the graph.

6. Conclusion not strong to present the finding. Please add data to support your conclusion. 

7. The discussion only describe the finding now compare the existing profile dimensionless concentration over the thickness of concrete.

Reviewer 2 Report

This is a well done study that models dynamics of mass transfer in underground structures. For figure 4, I recommend increasing the font size for the X and Y axis.

Reviewer 3 Report

In this paper, the authors conducted mathematical modeling to understand the concrete corrosion process. The structure-shore-water model involving parameters such as concentration and density is proposed. I have several questions and suggestions for this work, as commented in the following.

1. In the introduction, the author mentioned “It is advisable to apply mathematical modeling to solve the problems of this type…”. Is there any similar model in the literature? In addition, “Well-known reference data” is mentioned on page 6, please refer to relevant literature.

2. The references cited by the author are rather general, and many references are piled up at the beginning or end of the paragraph. Please revise the introduction.

3. On page 3, the author motioned “Let’s assume that the values of densities of soil and aquatic medium are different”. What does the aquatic medium refer to? Why is it different from soil density, why is this assumption necessary, i.e. will they ever have the same density? The author made several assumptions in this paper. The reasonableness and basis should be stated.

4. On page 3 and other pages, the author mentioned “so-called” several times. I think the word has a certain pejorative connotation and is not appropriate to be used here. The author should minimize colloquial expression in the manuscript, and an objective, concise, rigorous description is needed.

5. In the mathematical model, the materials are mainly described by concentration and density. How do the properties of materials like cement and soil correspond to the parameters in the model? For example, the calcium hydroxide, salinity properties.

6. All characters in the equations should be explained, such as q, k, β, etc. on page 3.

7. Figure 5 is unclear. Figures in the paper should be self-explanatory.

8. The mathematical model should be validated.

Reviewer 4 Report

My comments to the manuscript (materials-2336732: An engineering method of analyzing the dynamics of mass transfer during concrete corrosion processes in offshore structures) could be found below.

1.       Introduction is quite simple and poorly written. Objective is not found.

2.       Methodology is very long. However, it is not clear and is hard to follow.

3.       Results are presented confusing. It is hard to find what are the main results of this work. Discussion is not found.

4.       Conclusion is poorly written.   

5.       Finally, the novelty is not found in the present manuscript. Therefore, it could not be recommended to publish in the Materials.

Round 2

Reviewer 3 Report

The comments are well addressed.

Reviewer 4 Report

The manuscript entitled “materials-2336732: An engineering method of analyzing the dynamics of mass transfer during concrete corrosion processes in offshore structures” is revised. The authors tried to add some more information in the revised manuscript. However, methodology is hard to follow. Results are presented confusing. It is hard to find what are the main results of this work. Discussion is not found. Conclusion is poorly written. The novelty is not found in the present manuscript. Therefore, it could not be recommended to publish in the Materials.